# UV Sensor Based on Surface Acoustic Waves in ZnO/Fused Silica

**DOI:** 10.3390/s23094197

**Published:** 2023-04-22

**Authors:** Cinzia Caliendo, Massimiliano Benetti, Domenico Cannatà, Alessio Buzzin, Francesca Grossi, Enrico Verona, Giampiero de Cesare

**Affiliations:** 1Institute for Photonics and Nanotechnology (IFN), Unit of Rome, National Research Council (CNR), Via del Fosso del Cavaliere 100, 00133 Rome, Italy; enrico.verona@cnr.it; 2Institute for Microelectronics and Microsystems (IMM), Unit of Rome, National Research Council (CNR), Via del Fosso del Cavaliere 100, 00133 Rome, Italy; massimiliano.benetti@cnr.it (M.B.); domenico.cannata@cnr.it (D.C.); 3Department of Information Engineering, Electronics and Telecommunications, Sapienza University of Rome, Via Eudossiana 18, 00184 Rome, Italy; alessio.buzzin@uniroma1.it (A.B.); francesca.grossi@polimi.it (F.G.); giampiero.decesare@uniroma1.it (G.d.C.)

**Keywords:** ZnO, photoconductivity, acoustoelectric effect, fused silica, SAW, UV light, sensors

## Abstract

Zinc oxide (ZnO) thin films have been grown by radio frequency sputtering technique on fused silica substrates. Optical and morphological characteristics of as-grown ZnO samples were measured by various techniques; an X-ray diffraction spectrum showed that the films exhibited hexagonal wurtzite structure and were c-axis-oriented normal to the substrate surface. Scanning electron microscopy images showed the dense columnar structure of the ZnO layers, and light absorption measurements allowed us to estimate the penetration depth of the optical radiation in the 200 to 480 nm wavelength range and the ZnO band-gap. ZnO layers were used as a basic material for surface acoustic wave (SAW) delay lines consisting of two Al interdigitated transducers (IDTs) photolithographically implemented on the surface of the piezoelectric layer. The Rayleigh wave propagation characteristics were tested in darkness and under incident UV light illumination from the top surface of the ZnO layer and from the fused silica/ZnO interface. The sensor response, i.e., the wave velocity shift due to the acoustoelectric interaction between the photogenerated charge carriers and the electric potential associated with the acoustic wave, was measured for different UV power densities. The reversibility and repeatability of the sensor responses were assessed. The time response of the UV sensor showed a rise time and a recovery time of about 10 and 13 s, respectively, and a sensitivity of about 318 and 341 ppm/(mW/cm^2^) for top and bottom illumination, respectively. The ZnO/fused silica-based SAW UV sensors can be interrogated across the fused silica substrate thanks to its optical transparency in the UV range. The backlighting interrogation can find applications in harsh environments, as it prevents the sensing photoconductive layer from aggressive environmental effects or from any damage caused by cleaning the surface from dust which could deteriorate the sensor’s performance. Moreover, since the SAW sensors, by their operating principle, are suitable for wireless reading via radio signals, the ZnO/fused-silica-based sensors have the potential to be the first choice for UV sensing in harsh environments.

## 1. Introduction

ZnO is a well-known piezoelectric material that shows remarkable acoustic properties, such as large piezoelectric-coupling coefficient, strong acoustoelectric interaction, and ease of integration with electronic circuits. Highly c-axis-oriented ZnO thin films can be grown by many techniques, such as chemical vapor deposition, sol–gel, spray-pyrolysis, molecular beam epitaxy, pulsed laser deposition, vacuum arc deposition, and magnetron sputtering [1,2]. The sputtering method was the obvious choice for the present research, as it is a cheap, relatively fast and well-established technology. ZnO is an attractive material suitable for short-wavelength sensor applications; it is transparent to visible light and has a wide direct-energy band gap of approximately 3.3 eV at room temperature, a long photo-carrier lifetime, free exciton binding energy of about 60 meV and is highly resistant to radiation damage [3]. UV sensors find applications in many fields, such as space communication, biomedical instrumentation, high-temperature plasma research, military applications and missile launching and testing, to cite just a few applications [4]. ZnO is a semiconductor and a piezoelectric material at the same time; this unique characteristic allows the ZnO to recover the double role of UV-sensitive layer and surface acoustic wave (SAW) transducer. As a result, ZnO-based UV sensors can be fabricated which are based on the measure of the SAW characteristics perturbed by the incident UV radiation. The absorbed UV radiation causes the generation of electron-hole pairs in the ZnO layer; the interaction of the free carriers with the electric field accompanying the propagating SAW results in a reduction of the phase velocity and an increase of the propagation loss of the wave, which, in turn, represents the sensor output.

In the present paper, thin and thick c-axis-oriented ZnO layers were grown by radio frequency reactive magnetron sputtering technique on fused silica substrates; the films’ structural and morphological characteristics were investigated by X ray diffraction (XRD) and scanning electron microscopy (SEM). UV light absorption measurements were performed to estimate both the penetration depth of the optical radiation in the 200 to 480 nm wavelength range and the ZnO bandgap. SAW delay lines consisting of a pair of metal interdigital transducers (IDTs) were photolithographically implemented onto the ZnO surface to measure the SAW phase velocity in darkness and under UV illumination. The SAW devices were tested in the 100 to 450 mW UV power range and no saturation effects of wave velocity were observed.

It is worth mentioning that research in the field of UV SAW sensors started about 20 years ago, and since then it has continued, fueled by ever-new ideas for pursuing the goal of improving the sensors’ performance, for example, by exploring different thicknesses of the UV-sensing layer, combining ZnO nanowires or nanosheets with ZnO thin films [5,6], fabricating multilayer structures [7], or exploring harmonic modes [8,9]. The last paragraph compares the performance of the ZnO-based SAW UV sensors described in the available literature (such as working frequency, sensitivity, rise and recovery time, and tested UV power range) and outlines some possible solutions to be explored to improve the sensors’ performance.

## 2. ZnO Deposition and Characterization

### 2.1. ZnO Layer Deposition

ZnO thin films were deposited by radio frequency reactive magnetron sputtering (by Ionvac Process s.r.l.) on fused silica substrates, which are transparent to UV radiation and inexpensive. The SiO_2_ substrates were first cleaned in an ultrasonic bath with soapy deionized water to remove the impurities on the substrates’ surface, then rinsed in deionized water and blow-dried with nitrogen. Finally, they were cleaned in ethanol and acetone for 10 min. The clean substrate was mounted in the sputtering chamber and placed under the target (a 4″ diameter, high-purity (99.999%) zinc target) and onto the heated work holder, which was kept at a constant temperature of 200 °C during the deposition process. The sputtering chamber was evacuated, and the pre-sputtering process was carried out for at least 15 min in Ar (20 standard cubic centimeters per minute (sccm), working pressure 3 × 10^−3^ Torr) to clean the target surface and remove impurities. The ZnO layers were deposited onto fused silica substrates according to optimized sputtering parameters [10,11], which are listed in Table 1.

ZnO films of different thickness (grown for different sputtering times) were deposited onto fused silica and silicon substrates in the same sputtering run. The sputtering system was equipped with a quartz crystal microbalance (QCM) to control the film’s growth rate and thickness. To yield accurate results, the thickness was also measured by means of SEM photos of the cross-section of the films grown on Si substrate, which was easy to cleave without causing damage to the film.

### 2.2. Morphological Characterization

Morphological characterization and thickness measurements of the ZnO layers were performed by SEM (Zeiss Evo MA10). XRD measurements were performed to characterize the ZnO films’ orientation (Geiger-Flex X-ray diffractometer from Rigaku Co., Ltd. Tokyo, Japan).

Figure 1a shows a SEM photo of the ZnO film cross-section: the film shows a dense columnar structure, and the film’s surface appears crack-free and uniform. SEM photos were performed on the ZnO layers grown on Si because the latter can be cleaved for cross-sectional imaging more conveniently than can fused silica. The films are highly adhesive to the substrate; the adhesion strength of the films to the SiO_2_ substrate was assessed by a rudimentary tape test. Figure 1b shows the XRD pattern of a ZnO film (4 μm thick), as an example.

The diffracted intensities were collected by a Rigaku Geiger-Flex X-ray diffractometer with a θ–2θ type system configured in Bragg-Brentano geometry in the range 20–40° with a step size of 0.1°. The X-ray source was a copper tube with a characteristic emission line of 1.54 Å/8.047 keV (Cu-Kα1) and it operated at a power of 1.6 kW (40 kV × 40 mA). The rocking curve was collected with a step size of 0.05°. The film was purely oriented with the c-axis perpendicular to the growth plane, as demonstrated by the sharp peak at 34.5° corresponding to a c-axis orientation perpendicular to film plane (002). The full width half maximum (FWHM) of the rocking curve was about 2.9°, indicating a small dispersion of the crystallites and good quality of film crystalline structure. No other peaks of ZnO were visible.

### 2.3. Optical Characterization

The optical transmission of the ZnO thin films was characterized in the UV spectrum from 200 to 450 nm by using a Deuterium-Tungsten Halogen Light Source (Mikropack DH-2000) and a Horiba Jobin Yvon monochromator to evaluate the UV penetration depth at different optical wavelengths. Five ZnO/SiO_2_ samples were prepared (ZnO layer thicknesses equal to 100 nm, 150 nm, 200 nm, 250 nm, and 400 nm); the films’ thickness was measured by SEM photos. Preliminary transmittance measurements were performed on the bare fused silica sample to confirm its optical transparency in the spectrum of interest (between 250 nm and 450 nm). Then, the optical characterization was carried out on the ZnO samples: the optical power ($P_{out}$) from the monochromator and the optical power ($P_{T}$) transmitted through the 5 samples were collected to derive the optical absorption *α* of the layer by using the Beer-Lambert law $P_{T}=P_{out}\cdot e^{-\alpha t}$, where t is the layer thickness. Figure 2a shows the measured absorption coefficient *α* vs. optical wavelength curve; Figure 2b shows the light penetration depth (derived as 1/α) vs. the optical wavelength curve.

The absorption coefficient of the sputtered ZnO layer at 365 nm working wavelength was around $3.1\times 10^{6}$ m^−1^, while the penetration depth was about 325 nm. The ZnO bandgap energy can be estimated by following the Tauc method, under the assumption that the energy-dependent absorption coefficient *α* can be expressed as follows [12]:(1)$${\left(\alpha h\nu \right)}^{\gamma }=B(h\nu -E_{g})$$
where h is the Planck constant, ν is the photon’s frequency, $E_{g}$ is the band gap energy, and *B* is a constant. The γ factor was set as equal to 2 for direct transition band gaps. Figure 3 shows the plot of ${\left(\alpha h\nu \right)}^{2}vs.h\nu$, the Tauc plot: the *x*-axis intersection of the linear fit gives an estimate of the band gap energy, 3.362 eV which closely agrees with the 3.37 eV of the bulk material [13,14]; 366.38 nm is the optical wavelenth at which the absorption starts. From Figure 3a the characteristic features of Tauc plots are evident: at low photon energies the absorption approaches zero (the ZnO is transparent); near the band gap value the absorption becomes stronger and shows a region of linearity in this squared-exponent plot. This linear region is used to extrapolate to the *x*-axis intercept to find the band gap value (here about 3.362 eV).

The density of electronic states in valence and conduction band tails into the energy band-gap due to defects in the film microstructure; near the optical band edge (energy range E < $E_{g}$) the optical absorption coefficient *α* shows a tail (Urbach tail) which can be expressed by an exponential dependence on photon energy E=hν [15]:(2)$$\alpha =\alpha_{0}\cdot exp(\frac{E-E_{g}}{E_{u}})$$
where $\alpha_{0}$ is a constant which can be determined from the converging point of the ln(*α*) vs. E plot shown in Figure 3b. The Urbach energy, $E_{u}$ = 0.15 eV, is calculated as the inverse of the slope of the linear fit of $\ln\left(\alpha \right)vs.E$.

## 3. ZnO Based SAW Delay Lines

Aluminum IDTs were photolithographically patterned onto the ZnO’s free surface by conventional optical lithographic technique and lift-off process. The IDTs were electrically connected to a Keysight P9371A vector network analyzer to measure the scattering parameters of the SAW travelling along the surface of the ZnO/SiO_2_ substrate. All measurements were taken in ambient air at room temperature. A high-power LED (Ledmod.v2 from Omicron-Laserage Laserprodukte GmbH) at 365 nm was employed as UV source (maximum optical power 450 mW). The LEDMOD V2 module is delivered with the Omicron Control Center (OCC)—Control Software which configures the LEDMOD’s for computer independent operation via RS-232 and USB 2.0 interface.

The fundamental peculiarity of a SAW is that it travels along the surface of the propagating piezoelectric medium; both the mechanical and electrical fields are mostly trapped at a depth approximately equal to one wavelength from the surface. Due to the electric field’s oscillation at the sound’s wavelength, a layer of bound charges at the surface of the piezoelectric medium is formed. When the incident UV light is absorbed by the ZnO, electron-hole pairs are generated that interact with the electric field accompanying the propagating SAW, resulting in a phase shift and time delay across the SAW device. This phenomenon is called the *acousto-electric effect* (AE) and is induced by illuminating the ZnO layer with UV light at 365 nm which corresponds to the ZnO energy bandgap. The sensing mechanism of the SAW UV sensor based on ZnO/fused silica involves generation of photogenerated charge carriers under UV illumination that increase the conductivity of the ZnO layer. The increase in conductivity leads to a decrease in SAW velocity. According to the perturbation theory [16], the relative SAW phase shift (which is equal to minus the relative SAW velocity change) induced by the conductivity variations can be expressed by the following approximate formula:(3)Δvv0=−K2211+σcσs
where *K*^2^ is the electromechanical coupling factor for the SAW device, *σ_s_* = *σ* · *h* is the sheet conductivity of the layer, *h* and *σ* are the conductive layer thickness and bulk conductivity, $\Delta v=v-v_{0},$ $v_{0}$ and v are the SAW unperturbed and perturbed velocities,$\sigma_{c}=v_{0}c_{s}=v_{0}\left(\varepsilon_{0}+\varepsilon_{s}\right)$, and *ε*_0_ and *ε_s_* are the dielectric permittivity of air and of the piezoelectric half-space. According to Equation (3), it can be observed that, under UV irradiation, when the conductivity of the photoconductive layer increases, the SAW velocity decreases [17]. In the present research effort, the SAW phase changes induced by UV lighting are correlated to the incident light power density and represent the UV sensor output.

A ZnO/SiO_2_-based SAW delay line (DL) was fabricated, which consisted of two equal unidirectional IDTs, as shown in Figure 4a. One IDT, the transmitting one, launched the SAW and the other IDT, the receiving one, reveal the wave; the synchronism frequency of the DL was f=v/λ, with *v* as the wave phase velocity and *λ* the IDT periodicity. Each IDT consisted of 80 electrodes in split-finger configuration to minimize overall transducer reflection, with an electrode width of 10 μm and a periodicity of *λ* = 80 μm, as shown in Figure 4b. The acoustic aperture (the IDT fingers overlapping) is 1568 μm and the IDT’s center-to-center distance is L = 6600 μm.

Conventional photolithographic and lift-off techniques were employed to pattern the IDTs onto an Al layer (1500 Å thick) grown onto the ZnO layer (about 4 µm thick) by radio frequency sputtering technique from an Al target in an Ar atmosphere. The SAW DL was assembled on a printed circuit board (PCB), and the solder pads were electrically bounded to SMA connectors to measure the scattering parameter S_21_ (which represents the power transferred from the transmitting IDT toward the receiving one) vs. frequency curve by using a vector network analyzer (VNA) (Keysight P9371A). The operational frequency of the Rayleigh mode was measured in darkness (40.48437 MHz) and then the UV radiation was focused onto the center of the acoustic propagation path between the two IDTs by means of an optical fiber coupled to the LED. Figure 5 shows a schematic of the measurement arrangement: the VNA and the UV LED were controlled by a PC through USB connections and proper software; the SAW device was connected to the VNA through SMA connectors. The SAW DL was assembled on a printed circuit board drilled to allow back-lighting of the SAW propagation path. The PC was connected to a LAN to allow remote access to the VNA and LED; the remote user could run applications and manage the measurements by linking to the host computer via the internet. The UV light was focused onto the ZnO film surface, along the SAW propagation path, with the spot area equal to 0.02556 cm^2^ and 1.5 mm the distance between the fiber and the device surface. The fiber had a diameter of 1 µm and numerical aperture equal to 0.5; since the UV light spot was larger than the acoustic beam width, some UV power was lost.

The “phase shift” method was used to characterize the SAW UV sensor response; the phase (at constant frequency) of the signal at the output of the DL was measured at different UV power values and compared to the reference signal (the phase measured in darkness). The relative phase change resulting from the UV light absorption was equal to the negative relative velocity change of the SAW. The measurement setup of the SAW sensors was intentionally located inside a darkroom which was part of a clean room to prevent spurious effects from environmental parameters changes (such as temperature, relative humidity and light). Inside the laboratory (clean room laboratory class 10.000 rated) the air temperature (24 °C) was stable within 1 °C, and its relative humidity was maintained at (40 ± 2)%.

## 4. SAW UV Sensor

Figure 6 shows two different UV lighting conditions adopted in the present experiment: top lighting, when the source of UV light is placed above the ZnO/fused silica, and bottom lighting, when the optical source is below the ZnO/fused-silica.

The SAW phase at constant frequency, $\varphi =\frac{2\pi L}{\lambda }=2\pi Lf/v$, was monitored in time during darkness/UV-light/darkness cycles, where L is the wave path and *v* the phase velocity; any wave velocity change induced by UV light absorption corresponds to phase shifts. The sensor time response to UV power density changes was measured as the difference between the phase values in darkness and under UV light. Figure 7 shows the SAW sensor time response during nine UV cycles for top illumination and the optical power density from the LED source vs. the time curve. The inset shows a magnification of the SAW sensor time response under 10.9 mW/cm^2^ UV power density. Similar results were obtained for back-illumination. As expected, under UV illumination, the phase shift increased as consequence of the decreased SAW phase velocity. The SAW sensors phase response is reversible, repeatable, and quite fast (rise time 10 s and recovery time 13 s), and its magnitude increased with increases in the incident UV power. The response time was calculated as the time required for the sensor to reach 90% of the total response; the recovery time was calculated as the time required for the sensor to return to 90% of the unperturbed baseline signal upon removal the UV light.

The calibration curve, the SAW phase shift vs. UV power density curve, shows a linear behavior, as reported in Figure 8, where each point is the average of four independent measurements and standard deviations are indicated as error bars. For both front and back illumination, the relative standard deviations between responses obtained in the same conditions were within 10%, demonstrating a good repeatability of the system. The SAW sensor sensitivity S, the relative phase shift per unit UV power density, was calculated as the slope of the linear fit of the experimental data shown in Figure 8: S is 318 and 341 ppm/(mW/cm^2^) for front and back-illumination.

The sensitivity value for back illumination was estimated without considering the UV energy loss due to reflections at the air/SiO_2_ and SiO_2_/ZnO optical interfaces. According to the theoretical predictions of Ref. [18], it was demonstrated that the photoconductive ZnO induces a short-circuiting effect of the electric potential Φ associated with the SAW. As a result of the UV absorption from the bottom side, the electric potential of the Rayleigh wave is expected to be null at the quartz/ZnO interface, and concentrated inside the piezoelectrically active portion of the ZnO layer (not involved in the UV absorption) where it reaches a peak at the ZnO/air interface where the IDTs were located. Thus, a higher output signal (and hence a higher sensitivity) is expected for Rayleigh wave-based sensors under bottom illumination.

## 5. Discussion

During this research, we reported very encouraging results from our UV-sensing experiments, notably the sensing of UV light on ZnO thin films using SAW devices for two different UV illumination geometries. No features of the saturated tendency of the SAW sensor appeared in the measured UV power range, indicating that the ZnO is very suitable for practical applications. The response of our SAW-based UV sensor had a rise and decay time of about 10 and 13 s, respectively, and a sensitivity of about 341 and 318 ppm/(mW/cm^2^) for back and top illumination, respectively. The ZnO film was c-axis-oriented, had a dense columnar structure, and its band gap was close to that of the bulk material. Table 2 lists some experimental data referring to the performances of ZnO-based UV sensors, as described in the available literature.

A comparison between the performance of our SAW sensor and those reported in the literature is not simple to perform for four main reasons: (1) only a few published papers study the response of the SAW sensors for an UV power range, while most consider a single power value; (2) most of the published articles do not specify whether the whole SAW device rather than only the SAW propagation path or only the IDT area are illuminated during the sensors test; (3) most of the published articles do not give a detailed description of the experimental setup used (which should include the power of the UV source, the area of the UV light spot, the distance between the light source and the device surface); and (4) most of the published articles do not specify the rise and recovery time of the studied UV sensors. This information is fundamental to the complexity of the problem and necessary for it to be adequately analyzed in order to plan possible improvements in sensor performance. For example, the results set forth in Ref. [32] show that the response of the SAW sensor to the UV light focused on the IDTs was up to five times greater than the response obtained by illuminating the propagation path of the wave. This effect is attributed to the decrease in depletion-width at the metal-semiconductor junction under the IDT and it resulted in an increase in the IDTs’ capacitance. According to these experimental results, our sensor’s performance can be improved if the IDTs (instead of the SAW propagation path) are illuminated by UV light. In Ref. [30] the studied SAW sensors exhibited rise and recovery times (11 and 44 s) quite similar to those measured in our experiment. In Ref. [31], the rise and recovery times of three SAW UV sensors based on 128° yx-LiNbO_3_ substrate were studied for different UV-sensing layers, namely ZnO, Sn/ZnO, and Ag-nano wires (NW)/ZnO. Among the three sensing layers, the AgNW/ZnO-based sensor showed a significant improvement in response time (3 s as opposed to 250 and 37 s for the ZnO- and Sn/ZnO-based sensors) and in the recovery time (3 s as opposed to 400 and 298 s for the ZnO- and Sn/ZnO-based sensors).

Our sensor performances can be optimized in some ways: (1) by choosing the proper ZnO layer thickness and electrical boundary conditions able to ensure higher electroacoustic coefficient *K*^2^ and hence higher sensitivity (according to Equation (3)); (2) by illuminating the IDTs implemented onto the ZnO layer’s surface [32]; and (3) by using glass (instead of fused silica) to reduce the production cost, even if at the price of losing the back-surface interrogation.

It is opinion of the corresponding author that the UV sensitivity of ZnO-film-based SAW devices is a challenging issue for study with many hitherto unexplored design aspects. Further theoretical and experimental studies are necessary to optimize the SAW sensor’s performance; substrate type, ZnO layer thickness, electroacoustic coupling configuration and acoustic mode are some of the design parameters to be investigated to enhance the sensitivity of the sensors based on the acoustoelectric effect.

It is worth mentioning that ZnO/SiO_2_ substrate-based acoustic wave sensors can be designed for [33,34] sensing a liquid’s properties, such as viscosity, density and mass anchored to the sensor surface. Therefore, a sensing platform based on ZnO/fused silica can be designed for the simultaneous measurement of multiple parameters related to both gaseous and liquid environments.

## 6. Conclusions

UV sensors find applications in many different fields, including pharmaceuticals, automobiles, and the chemical industry (for the production-storage-transportation of chemicals), just to cite a few. UV sensors must be fast, inexpensive, highly sensitive, long-lasting, rugged, and easy to clean to minimize errors caused by dust that can block the path of the radiation to be detected. Ideal sensors should be robust, and their electronics should not be exposed to external stimuli to avoid being damaged by the surrounding environment. SAW sensors meet these requirements; their characteristics include low power consumption, light weight, functional versatility, and low cost; moreover, due to their frequency readout, they are suitable for remote control and wireless sensing in hostile environments.

The ZnO/fused silica-based SAW UV sensor response, i.e., the wave velocity shift due to the acoustoelectric interaction between the photogenerated charge carriers and the electric potential associated with the acoustic wave, was measured for different UV light power densities focused along the wave propagation path, under illumination from the top surface of the ZnO layer and from the fused silica/ZnO interface. The reversibility and repeatability of the sensor responses were assessed. The time responses of the UV sensor showed a rise time and a recovery time of about 10 and 13 s, respectively, and a sensitivity of about 318 and 341 ppm/(mW/cm^2^) for top and bottom illumination, respectively. Our sensing device, whose performance is comparable to those of sensors based on more complex structures or fabricated with more sophisticated technologies, is based on a simple bilayer configuration and cheap fabrication method, and thus can be considered competitive in the UV sensors landscape. The backlighting of the fused silica/ZnO substrate can find applications in harsh environments, as it prevents the sensing photoconductive layer from aggressive environmental effects or from any damage caused by cleaning the surface from dust which could deteriorate the sensor’s performance.

Before the ZnO-based SAW UV sensors can be transitioned from laboratory proof-of-concept to commercial devices, further theoretical studies and experiments are needed to optimize the design parameters (such as substrate type, sensing layer thickness and acoustic mode) and enhance the performance of the ZnO-based UV sensors, which are demonstrating their claim to be research-wise in the field.

## Figures and Tables

**Figure 1 sensors-23-04197-f001:**
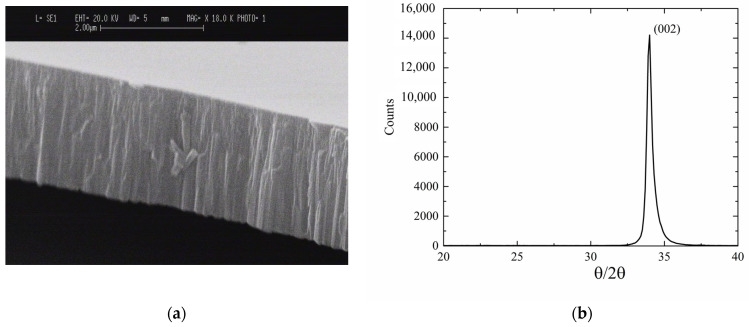
(**a**) The SEM photo of the cross section of the ZnO layer grown on Si substrate; and (**b**) the XRD pattern of the ZnO layer grown on fused silica.

**Figure 2 sensors-23-04197-f002:**
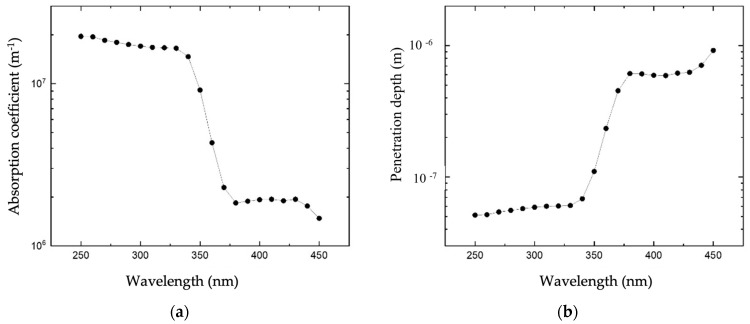
(**a**) Absorption coefficient of the sputtered ZnO thin film vs. wavelength. (**b**) Penetration depth of photons inside the sputtered ZnO thin film vs. photons’ wavelengths.

**Figure 3 sensors-23-04197-f003:**
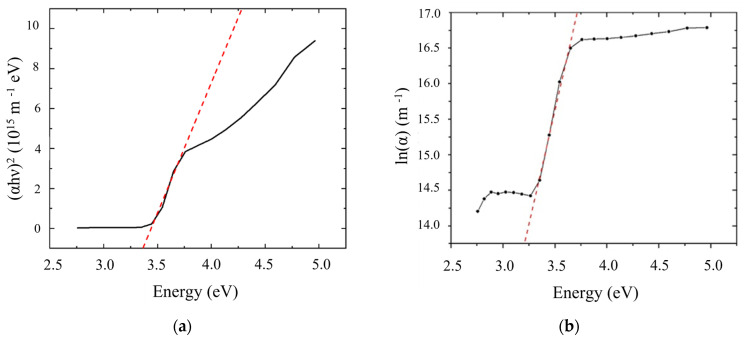
(**a**) the Tauc plot of the ZnO samples which illustrates the method of fitting the linear region to evaluate the bandgap at the photon-energy axis intercept, here about 3.362 eV; and (**b**) the $\ln\left(\alpha \right)vs.h\nu$ plot used to calculate the Urbach energy.

**Figure 4 sensors-23-04197-f004:**
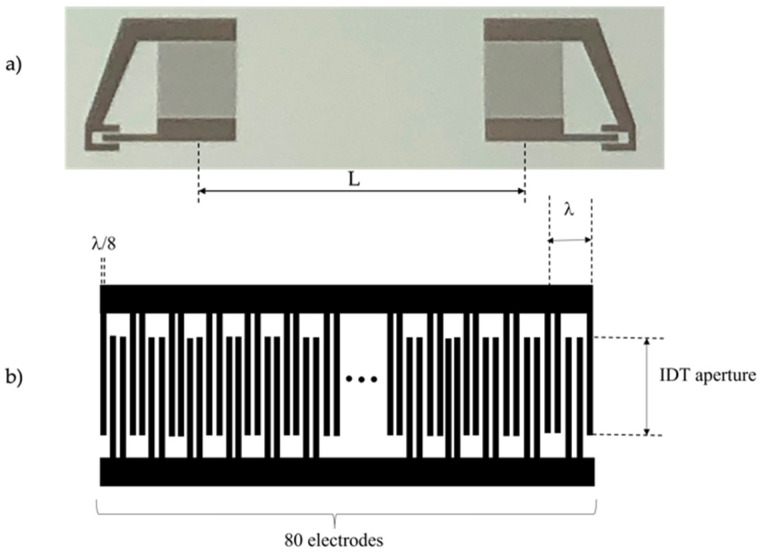
(**a**) The photo of the SAW delay line consisting of two IDTs; and (**b**) the schematic of the unidirectional split-finger IDT.

**Figure 5 sensors-23-04197-f005:**
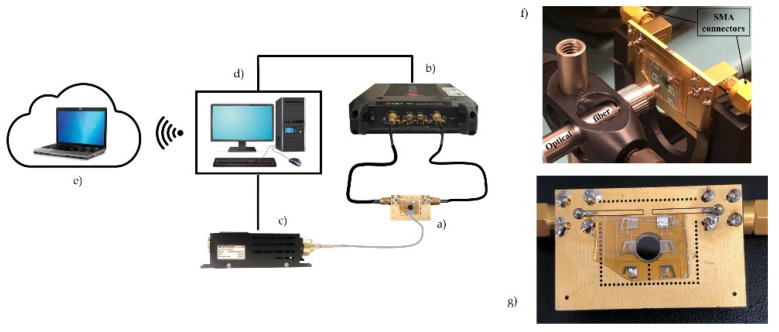
The schematic of the measurement setup: (**a**) the SAW device; (**b**) the network analyzer; (**c**) UV led driver; (**d**) data acquisition system; (**e**) remote control; (**f**) an enlarged photo of the SAW sensor and of the fiber; and (**g**) an enlarged photo of the SAW DL assembled on the printed circuit board drilled to allow rear lighting.

**Figure 6 sensors-23-04197-f006:**
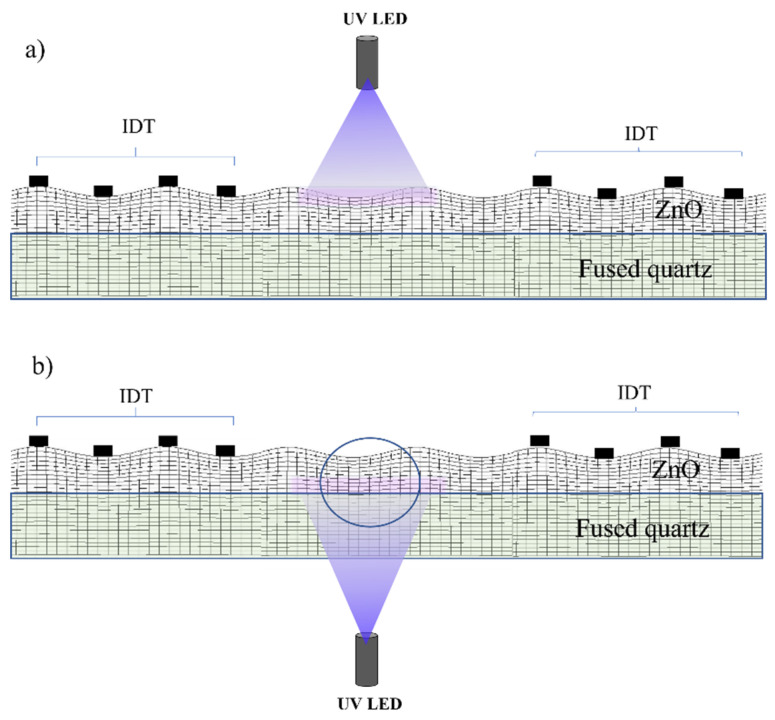
The schematic of the SAW sensor for (**a**) top and (**b**) bottom illumination. The picture is not in scale.

**Figure 7 sensors-23-04197-f007:**
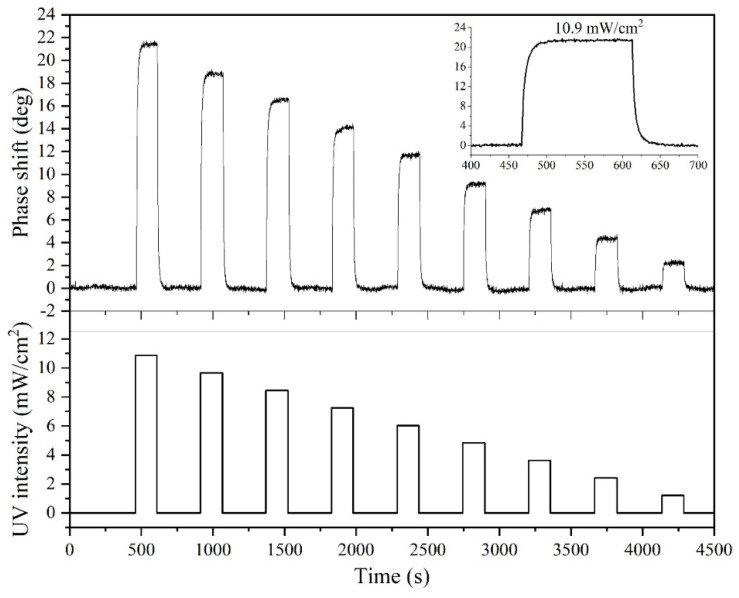
The time response of the SAW sensor for top illumination and the optical power from the LED source vs. time curves. The inset shows a magnification of the SAW sensor time response under 10.9 mW/cm^2^ UV power density.

**Figure 8 sensors-23-04197-f008:**
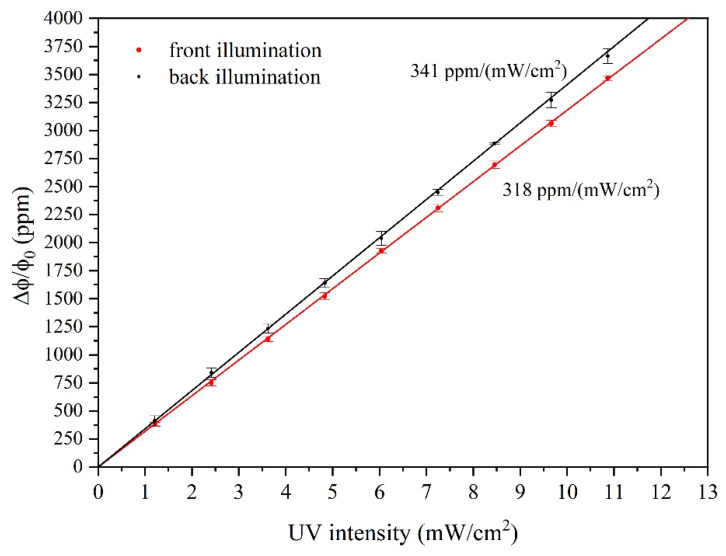
The relative phase shift vs. the UV power density for top and bottom illumination.

**Table 1 sensors-23-04197-t001:** The deposition parameters of the ZnO layers.

Radio frequency power	200 W
Substrate temperature	200 °C
Gas residual pressure	7 × 10^−7^ Torr
Gas working pressure	3.7 × 10^−3^ Torr
Gas composition	Argon/Oxygen in 1:4 ratio
Gas purity	99.9999%
Thickness	from 100 nm to 4 μm

**Table 2 sensors-23-04197-t002:** The SAW sensor type, the resonant frequency, the frequency shift, the incident UV power, and the corresponding reference.

SAW UV Sensor Device	Frequency (MHz)	Frequency Shift or Relative Frequency Change	UV Power (mW/cm^2^) at 365 nm	Ref.
ZnO(200 nm)/LiNbO_3_	37	170 kHz	40	[19]
ZnO(71 nm)/LiNbO_3_	35	28 kHz	0.034	[20]
ZnO(200–400 nm)/Mg:ZnO/ZnO(2 µm)/sapphire	711.3	1360 kHz	2.32	[21]
ZnO(3.23 µm)/Si	842.8	1017 kHz	0.551	[7]
ZnO(~1.5 µm)/36° Y LiTaO_3_	41.5	No significant frequency shift *	0.570	[22]
ZnO(250 nm)/128° yx LiNbO_3_	439	6000 ppm/(μW/cm^2^)	0.010 to 40	[23]
ZnO(70 nm)/silica	41.2	45 kHz	19	[24]
ZnO(500 nm)/Si	122.15 339.9	400 kHz (3rd harmonic) 10 kHz (fundamental)	3	[25]
ZnO(2 µm)/Corning glass 2318	211.5	70 kHz	7.6	[26]
ZnO(4–6.5 μm)/Al foil (160 μm) S0 Lamb mode	30	53.7 ppm/(mW/cm^2^)	2–25	[27]
ZnO(400 nm)/ST-quartz	196	35 kHz 50 kHz (saturation)	<16 16–300	[28]
ZnO(2.5 µm)/LiNbO_3_	107.98	5–23 kHz	1.8–4 µW/cm^2^	[29]
ZnO/SnO_2_/Ta_2_O_5_/LiNbO_3_	284	2.05 kHz	0.06	[30]
ZnO/128° yx LiNbO_3_	242.625	100 MHz	3.916 mW/cm^2^	[31]
Sn/ZnO/128° yx LiNbO_3_	241.875	58 kHz		
AgNW/ZnO/128° yx LiNbO_3_	242.25	68 kHz		
ZnO/fused silica	40.48	318 and 341 ppm/(mWcm^−2^) for top and bottom illumination	Up to 10.9	Present paper

* only insertion loss changes were observed.

## Data Availability

Not applicable.

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
