# Peer review of "UV Sensor Based on Surface Acoustic Waves in ZnO/Fused Silica"

_sensors, 2023, doi:10.3390/s23094197_

Round 1

Reviewer 1 Report

See the file attached.

Author Response

Dear Referees

First of all I would like to thank you for your comments and suggestions on how to improve my paper: I really appreciate your efforts. I did my best to correct the paper.

I used different colors for the revisions done according to the comments from each of you.

Referee n. 1

(1) Figure 1b: What is the x-axis unit? In general, the x-axis is normally labeled “2θ (degree)”.

The x-axis unit was added.

(2) You mentioned that “…the IDTs center-to-center distance is 6600 μm.” Could you please

show this dimension in Fig. 5 inset which is the total view of the IDT structure?

The dimension L was shown in the figure.

(3) Fig.6 is the photo of the experimental setup. Please add a schematic to show the overall

measurement arrangement to show how the vector network analyzer and other equipment are

connected to the device. Also, you used “VNA” in Fig. 6. Please define this acronym right after “vector network analyzer” in the text.

A schematic of the set up was added in the figure 5.

The acronym was added.

(4) Please use a few sentences to describe the novelty and new discoveries of this study.

A comparison between the performance of our SAW sensor and that of the literature is not simple to do for two main reasons: 1. only a few published papers study the re-sponse of the SAW sensors for an UV power range but rather for a single power value. 2. most of the published articles do not specify whether the whole SAW device rather than the SAW propagation path or only the IDT area are illuminated during the sensors test. This two information are fundamental for the complexity of the problem to be adequately analyzed. In particular, the latter information is crucial to evaluate possible improvements in sensor performance. To this end, the results set forth in reference 27 show that the re-sponse of the SAW sensor to the UV light focused on the IDTs is up to five times greater than the response obtained by illuminating the propagation path of the wave. This effect is attributed to the decrease in depletion width at the metal- semiconductor junction under the IDT and it results in an increase in the IDTs capacitance. According to these experi-mental results, our sensor performance can be improved if the IDTs (instead of the SAW propagation path) are illuminated by UV light.

(5) Because ZnO thin films are normally hydrophilic, any changes in environmental conditions such as humidity will significantly influence the detection precision and sensitivity of SAWbased UV sensors. Have you tested your device under different humidities?

The relative humidity was fixed. The following text was added to the manuscript:

 The measurements set up of the SAW sensors is intentionally located inside a darkroom which is part of a clean room to prevent spurious effects from environmental parameters changes (such as temperature, relative humidity and light). Inside the laboratory (clean room laboratory class 10.000 rated) the air temperature (24°C) is stable within 1 °C and its relative humidity is maintained at (40 ± 2)%.

(6) The detector response time usually refers to the time required for the signal to rise from 10% to 90% of the peak value and then fall from the 90% to 10% of the peak value under light irradiation. How were the rise and recovery times defined in the manuscript?

The response time was calculated as the time required for the sensor to reach 90% of the total response of the signal; the recovery time was calculated as the time required for the sensor to return to 90% of the unperturbed baseline signal upon removal the UV light.

(7) It is generally believed that differences in the surface defect concentration caused by

annealing will lead to changes in the response time for SAW-based UV sensors. Have you tried using annealing temperature control to improve the detector response speed?

Our ZnO films were grown at 200°C and no annealing was done.

(8) Since slow response speeds and high noise levels are the two main drawbacks of SAW-based UV detectors, any suggestions to improve the signal-to-noise ratio of the UV sensor?

As to my opinion the type of substrate and the ZnO film quality largely affect the signal to noise ratio; as to my experience, the fused quartz substrate was preferable to the silicon substrate due to noise in the SAW sensor response.

(9) Table 2 lists the performances of some reported SAW-based UV detectors. Please include the performance of this device in the same table.

The performances of our device were added to the list.

Thank you again

Best regards

Cinzia Caliendo

Reviewer 2 Report

This paper meant to proposed a ZnO based UV sensor. However, I do not think this work is good enough for publishing in a peer-reviewed journal.

1. The paper lacks of novelty. The ZnO based SAW UV sensor has been reported about 20 years ago. The submitted work shows no obvious improvement neither in theory nor technique.

2.The English writing is another problem to show the value of the paper.

3.The response time of tens of seconds is not an acceptable value for many applications.

4.No detailed photo for the finial device is shown.

5. The purpose of Fig.4 is vague. I do not think it is a necessary work to test the band-gap energy for the sensor.

6. Fig.1a is not on a SiO2 substract?

7. The introduction is poor to show the previous works existing problems and your contributions. The abstract also needs a promotion.

Author Response

Dear Referees

First of all I would like to thank you for your comments and suggestions on how to improve my paper: I really appreciate your efforts. I did my best to correct the paper.

I used different colours for the revisions done according to the comments from each of you.

Review n. 2

This paper meant to proposed a ZnO based UV sensor. However, I do not think this work is good enough for publishing in a peer-reviewed journal.

  1. The paper lacks of novelty. The ZnO based SAW UV sensor has been reported about 20 years ago. The submitted work shows no obvious improvement neither in theory nor technique.

The ZnO-based SAW UV sensor is recently renewed as demonstrated by the large number of papers published in the last years which demonstrate that not all has been explored in the field and that much still need to be investigated. The present research receives European funds from Lazio Region and the proposed research has been reviewed by a committee who decided that the topic of the research is eligible for funding. Moreover, a comparison between the performances of our SAW sensors and those referred to the SAW sensors described in the available literature is not simple to do since only a few published results calculate the sensor response for a small UV power range but rather for a single power value.

2.The English writing is another problem to show the value of the paper. English very difficult to understand/incomprehensible.

I am very sorry of your very bad opinion on my English!

The paper has been corrected by a native English colleague.

3.The response time of tens of seconds is not an acceptable value for many applications.

The response and recovery times of our SAW UV sensors are not so short to be suitable for ALL applications but: 1. are lower than those reported in many published papers; 2. Can be improved by illuminating the IDTs instead of the SAW propagation path.

4.No detailed photo for the finial device is shown.

 I am sorry but the paper includes a photo of the SAW device. 

  1. The purpose of Fig.4 is vague. I do not think it is a necessary work to test the band-gap energy for the sensor.

In the ZnO-based SAW UV sensors field the film bandgap plays a fundamental role as it controls the absorption and conductivity phenomena which are the basic phenomena of the SAW UV sensors. Moreover, a ZnO film with a bandgap energy close to the value of the bulk single crystal ZnO confirms that the quality of the ZnO is good and can be successfully used to fabricate SAW devices and SAW UV sensors. Some comments have been added to the figure as asked by another referee so now the need to show this figure is clearer than before.

  1. Fig.1a is not on a SiO2 substract?

During each sputtering deposition run two substrates were used, fused silica and Si. SEM photo was done on the ZnO film growth onto the Si substrate since the Si can be easily cut and the cross section has a good appearance without producing  damage to the film.

  1. The introduction is poor to show the previous works existing problems and your contributions. The abstract also needs a promotion.

The abstract and the introduction have been improved.

Thank you again

Best regards

Cinzia Caliendo

Reviewer 3 Report

This manuscript mainly focused on the preparation of  acoustic waves in ZnO/fused silica for UV sensors application. Some questions have arisen and the quality of this paper should be improved. I suggest the manuscript be published after revision. I have the following questions.

1. It is recommended that the author explain the mechanism of the sensor in detail.

2. Please describe the service life of the UV sensor.

3. The authors should state the experimental conditions, such as how high the room temperature is.

4. Ask the authors to explain how the thickness of the five samples was determined.

Author Response

Dear Referees

First of all I would like to thank you for your comments and suggestions on how to improve my paper: I really appreciate your efforts. I did my best to correct the paper.

I used different colours for the revisions done according to the comments from each of you.

Review n. 3

(x) English language and style are fine/minor spell check required

The English was improved with the help of a Mother English language colleague.

  1. It is recommended that the author explain the mechanism of the sensor in detail.

Some text has been added to explain the acoustoelectric effect which is the basic phenomenon of the SAW UV sensors.

  1. Please describe the service life of the UV sensor.

I am sorry but I am not sure to have understood the question by I try the same to answer it. The ZnO thin film technology is long time optimized process in my laboratory: samples as old as few years were interrogated and they respond as well to UV light (I am referring to ZnO based devices which were used for other applications). As to my experience, the ZnO films grown by rf magnetron sputtering are very stable, highly adessive to the substrate and and crack-free. We did no perform any aging tests on the presently studied devices.

  1. The authors should state the experimental conditions, such as how high the room temperature is.

The following text was added (coloured in yellow since the same question was from another Referee):

 The SAW sensors are interrogated remotely since both the UV led and the VNA are connected to a pc. The measurements set up of the SAW sensors is intentionally located inside a darkroom which is part of a clean room to prevent spurious effects from environmental parameters changes (such as temperature, relative humidity and light). Inside the laboratory (clean room laboratory class 10.000 rated) the air temperature (24°C) is stable within 1 °C and its relative humidity is maintained at (40 +- 2)%. No thermal drifts were observed.

  1. Ask the authors to explain how the thickness of the five samples was determined.

The thicknesses were measured by SEM photos of the layers cross section to confirm the readout from the quartz crystal microbalance which is put inside the sputtering chamber.

Thank you again

Best regards

Cinzia Caliendo

Reviewer 4 Report

The manuscript UV sensor based on surface acoustic waves in ZnO/fused silica reports that Zinc oxide (ZnO) thin films have been grown by the rf sputtering technique onto fused silica substrates. The authors measured the optical and morphological characteristics of the as-grown ZnO samples by various techniques such as X-ray diffraction spectrum, scanning Electron Microscopy, and light absorption measurements, which allowed the authors to estimate the penetration depth of the optical radiation in the 200 to 480 nm wavelength range and the ZnO band-gap. The wave velocity shift due to the acoustoelectric interaction between the photogenerated charge carriers and the electric potential associated with the acoustic wave was also measured for different UV power densities.

1.     This is a good work, the figures and references listed have been cited in the contents, and the discussion of related work and associated references are adequate.

2.     The paper is well-structured and written. The Abstract and the Introduction sections provide helpful information for the readers.

3.     The English is accepted. However, it can be improved, there are a lot of typos e.g

a.      ZnO Deposition and Charactetization

b.     In the conclusions a table is shown should be “In the conclusions, a table is shown

c.      During this research we should be “During this research, we…….

4.     In the figure captions and the figures in the text should be consistent. i.e, either figure or Figure and (a) or a)

5.     In Figure 4. The authors used the plot to get the estimated value of the band gap. This is used for only direct band gap semiconductors However, the authors did not discuss the Urbach tail for determining the band gap. The author can discuss this point or touch upon the determination and the error in getting the band gap value using this approach.

6.     The conclusion is very long and could be more convenient. It is shortened and compact, and the comparison part or discussion should be added to the end of the results and discussion section.

Author Response

 Dear Referee

Thank you for your comments on my paper I have corrected following your suggestions.

  1. This is a good work, the figures and references listed have been cited in the contents, and the discussion of related work and associated references are adequate.

Thank you

  1. The paper is well-structured and written. The Abstract and the Introduction sections provide helpful information for the readers.

Thank you

  1. The English is accepted. However, it can be improved, there are a lot of typos e.g 
  2. ZnO Deposition and Charactetization
  3. In the conclusions a table is shown should be “In the conclusions, a table is shown
  4. During this research we should be “During this research, we…….

These and other typos were corrected.

  1. In the figure captions and the figures in the text should be consistent. i.e, either figure or Figure and (a) or a)

 I corrected them.

  1. In Figure 4. The authors used the plot to get the estimated value of the band gap. This is used for only direct band gap semiconductors However, the authors did not discuss the Urbach tail for determining the band gap. The author can discuss this point or touch upon the determination and the error in getting the band gap value using this approach. 

 The Urbach tail was calculated.

  1. The conclusion is very long and could be more convenient. It is shortened and compact, and the comparison part or discussion should be added to the end of the results and discussion section. 

The “Discussion and conclusion” paragraph was divided in two paragraphs now entitled “Discussions” and “Conclusions”.

The  main advantages of the presently developed sensor are described in comparison with the known ones.

Thank you

Best regards

Cinzia Caliendo

Reviewer 5 Report

The topic is well known and studied from the mid 1070's. Of course the repetition of the research from a slightly different point of view is valuable itself. The paper is quite well-written, however I found a few drawbacks:

line 11 RF (like in line 66)

line 74 sccm (standard cubic centimeters per minute) the acronym is not well-known and should be ful named.

Table 1. Oxygen

Figure 2. is trivial and unnecessary

line 134 and 135 left hand side and right hand side greek nu are inserted using different fonts. Different font is used also in line 135. The same for g in the subscript.

It is necessary to comment the Figure 4. The chapter should not be finished with a drawing without any remark.

line 143 IDT means Interdigital Transducer not interdigitated electrodes

line 166 centre frequency (or synchronism frequency) it was delay line used not resonator

line 177 RF

Figure 6. remains lonely without any final remark (like Figure 4.) The letters in the description shuold be suited to the fonts in the text

Figure 7. Similar to the previous image the letters are disproportionately big 

line 223 and 233 The term UV adsorption is in my opinion wrong it would be better to use absorption.

Author Response

Dear Referees

First of all I would like to thank you for your comments and suggestions on how to improve my paper: I really appreciate your efforts. I did my best to correct the paper.

I used different colours for the revisions done according to the comments from each of you.

Review n. 4

  • line 11 RF (like in line 66)

radio frequency was written instead of RF.

  • line 74 sccm (standard cubic centimeters per minute) the acronym is not well-known and should be ful named.

The full name was writte.

  • Table 1. Oxygen

Correction done.

  • Figure 2. is trivial and unnecessary

Figure 2  has been deleted and all the figures have been renumbered.

  • line 134 and 135 left hand side and right hand side greek nu are inserted using different fonts. Different font is used also in line 135. The same for g in the subscript.

Corrections done.

  • It is necessary to comment the Figure 4. The chapter should not be finished with a drawing without any remark.

A comment was added after the figure.

  • line 143 IDT means Interdigital Transducer not interdigitated electrodes

correction done.

  • line 166 centre frequency (or synchronism frequency) it was delay line used not resonator

 correction done.

  • line 177 RF

radio frequency was written instead of RF.

  • Figure 6. remains lonely without any final remark (like Figure 4.)

A comment was added after the figure.

  • The letters in the description shuold be suited to the fonts in the text

Correction done.

  • Figure 7. Similar to the previous image the letters are disproportionately big

 The picture was corrected.

  • line 223 and 233 The term UV adsorption is in my opinion wrong it would be better to use absorption.

“absorption” was written along the text.

Thank you again

Best regards

Cinzia Caliendo

Round 2

Reviewer 1 Report

Thank you for your revision that has improved the quality of the manuscript. I have updated my recommendation for acceptance for publication.

Wishing you the best in your future research. 

Author Response

Dear Referee

thank you for your time and efforts dedicated to review the present paper.

and thank you also for your suggestion to publish the paper as is.

best regards

Cinzia Caliendo

Reviewer 2 Report

Some of my concerns have been treated, but some issues still exist.

1. I get the photos in Fig.5. However, the details of fabricated sensor cannot be found from the so-called enlarged figure. More detailed photo for the finial device (e.g. SEM or optical microscope photos) are needed to show the key elements.

2.The authors claimed that there are some applications without high responding/recovering speeds. Please named them in the discussion section.

3. A column for responding/recovering speeds should be added in Table 2.

4. I’m also wondering the measuring selection of this sensor. What about the output of your device when some other lights appear? It should be tested. 

Author Response

Response to Referee n. 2

  1. I get the photos in Fig.5. However, the details of fabricated sensor cannot be found from the so-called enlarged figure. More detailed photo for the finial device (e.g. SEM or optical microscope photos) are needed to show the key elements.

The photo of the device was added in figure 5.

2.The authors claimed that there are some applications without high responding/recovering speeds. Please named them in the discussion section.

The phrase “there are some applications without high responding/recovering speeds” is written nowhere inside the manuscript.

In the conclusion it is clearly written that :” Before the ZnO-based SAW UV sensors be transitioned from laboratory proof-of-concept to commercial devices, further theoretical studies and experiments are needed to optimize the design parameters (such as substrate type, sensing layer thickness and acoustic mode) and enhance the performance of the ZnO-based UV sensors which are demonstrating to be research-wise in the field.”

  1. A column for responding/recovering speeds should be added in Table 2.

Table 2 lists the resonant frequency, the frequency shift, the UV power, the sensitivity of some SAW UV sensors described in the available literature (the corresponding references are listed in the last column). The reason why the rise and/or recovery times of the studied UV sensors were not mentioned in the table is that about half of the papers listed do not give these values.

It seems to me better not to add a column with so many “not available” labels. Two references and some text (green coloured) have been added.

  1. I’m also wondering the measuring selection of this sensor. What about the output of your device when some other lights appear? It should be tested.

the SAW response is not affected by the light in the room but we preferred to perform our measurements in dark.

Works are in progress to test the response of our sensor to other optical wavelengths.

thank you

best reghards

Cinzia Caliendo
